# EGH-Net: Energy-Guided Hypergraph for Two-View Correspondence Learning

## Abstract

Learning reliable correspondences (inliers) and removing unreliable correspondences (outliers) is a fundamental task in computer vision. However, previous works based on local neighborhood graphs fail to effectively capture high-order constraints among nodes. To address these, we propose an Energy-Guided Hypergraph Network (EGH-Net), which leverages energy functions to guide the hypergraph to accurately capture higher-order constraints, thereby achieving more effective outlier rejection. Specifically, we first construct the hypergraph to capture group-wise relations, and then design the intra-graph energy function to compute feature differences among multiple nodes within subgraphs and to model the consistency constraints within hyperedges. Then, we design the inter-graph energy function to capture structural similarity across subgraphs, and implement it through the proposed Graph Kernel (GK) module using multi-scale feature decomposition. Finally, we optimize both intra- and inter-graph energy terms via stochastic gradient descent (SGD) to dynamically update feature representations, so as to improve the local geometric consistency of node features and effectively achieve structural alignment across subgraphs. Extensive experiments demonstrate that EGH-Net achieves superior performance compared to state-of-the-art methods across various visual tasks. *The code will be released when the paper is accepted.*

## 1 Introduction

Accurate identification of reliable correspondences between image pairs is fundamental for a wide range of vision tasks, such as 3D reconstruction (Wang et al., 2017), visual localization (Sattler et al., 2018), image registration (Wu et al., 2021), and *etc.* However, due to the presence of occlusion, viewpoint diversity, and structural ambiguities in complex scenarios, initial correspondences inevitably contain a significant number of outliers, which pose challenges for downstream tasks.

Thus, graph neural networks (GNNs) have been widely adopted to reject outliers. Representative methods, such as CLNet (Zhao et al., 2021), MS$^2$DGNet (Dai et al., 2022), NCMNet (Liu & Yang, 2023), MGNet (Luanyuan et al., 2024), and BCLNet (Miao et al., 2024), typically construct local neighborhood graphs using K-nearest neighbors (KNN) to model the local geometric relationships between nodes. However, these methods typically rely on pairwise relationships propagated within the graph structure, and the constructed neighborhood graph is typically restricted to local connectivity, making these methods difficult to model more complex higher-order constraints. For example, let us consider a challenging scenario in Fig. 1(a), where previous methods may fail to capture high-order geometric structural relationships, such as the coplanarity among nodes $a$, $c$, and $e$, or the local consistency among nodes $g$ and $i$ forming the corner points of a building window.

To this end, we propose to use hypergraphs to model higher-order constraints. Specifically, hypergraphs use hyperedges to simultaneously associate multiple nodes. In this way, they represent group-wise relations beyond simple pairwise connections and are thus suitable for modeling higher-order constraints. However, in practice, randomly distributed outliers are often incorporated into hyperedges, which disrupts local consistency and weakens the reliability of higher-order constraint modeling. Thus, how to enable hypergraphs to more accurately capture higher-order constraints without being disturbed by outliers presents a significant challenge.

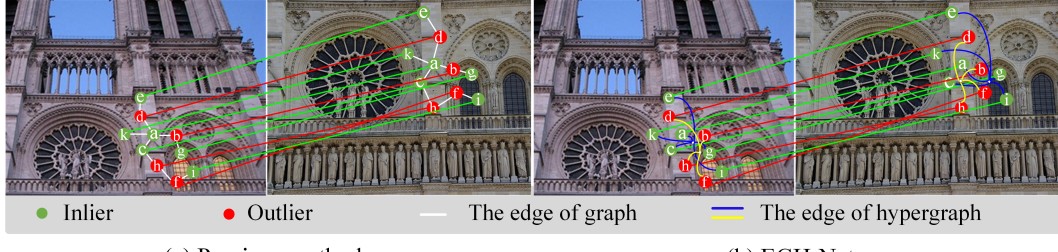

(a) Previous methods          (b) EGH-Net

Figure 1: (a) Previous methods are restricted to pairwise edges modeling, which may fail to capture higher-order constraints such as the coplanarity among nodes $a$, $c$, and $e$, or the corner structure formed by nodes $g$ and $i$. (b) In comparison, EGH-Net employs an energy function to guide the hypergraph in capturing higher-order relationships, thereby successfully modeling such constraints.

To address this challenge, we propose the Energy-Guided Hypergraph Network (EGH-Net), where we design two energy functions, inter- and intra-graph functions, to guide the hypergraph to better capture higher-order constraints, and then update node features to effectively suppress outliers and enhance inlier relationships, as shown in Fig. 2. Specifically, we first construct the hypergraph, and then propose a intra-graph energy function to compute the feature differences among multiple nodes within subgraphs. A low value of the inter-graph energy indicates that nodes within the subgraphs satisfy the consistency constraints, while a significantly high value often represents outliers that violate the consistency constraints. Thus, the intra-graph energy function can effectively model the consistency constraints within hyperedges. Then, to capture structural similarity across subgraphs, we design the Graph Kernel (GK) module, which computes inter-graph energy through multi-scale feature decomposition to align structural relationships. In the GK module, a bottom-up path captures the global structural features of subgraphs, while a top-down path preserves their local details. These features are then fused to adaptively model structural similarity across subgraph dependencies. Finally, EGH-Net combines intra- and inter-graph energy functions to update node features via gradient descent to jointly optimize consistency and structural alignment across subgraphs, thereby modeling high-order constraints and facilitating effective outlier rejection As shown in the aforementioned example of Fig. 1(b), the EGH-Net can successfully capture the coplanarity among $a$, $c$, and $e$, and the local consistency of $g$ and $i$ at a window corner.

Our contributions are summarized as follows.

(1) We propose the EGH-Net, an energy-guided hypergraph network designed for two-view correspondence learning. By capturing higher-order constraints and optimizing node features via gradient descent, EGH-Net effectively suppresses outliers and reinforces inlier relationships.

(2) We propose an intra-graph energy function to capture consistency constraints within subgraphs, and design a GK module to compute inter-graph energy to capture structural similarity and dependencies between subgraphs through a multi-scale bidirectional mechanism.

(3) EGH-Net achieves outstanding performance across multiple tasks. For instance, in the camera pose estimation task, EGH-Net achieves 4.76% and 4.67% improvements in the mAP5° for known and unknown outdoor scenes in the YFCC100M dataset, respectively, compared to the baseline, while reducing the number of training parameters by 37.56%.

## 2 RELATED WORK

### 2.1 OUTLIER REJECTION

Traditional RANSAC (Fischler & Bolles, 1987) and its variants (Barath et al., 2019; Chum et al., 2005) iteratively perform random sampling of the minimal point set to fit a model and retain inliers that are consistent with the model, thereby removing outliers that do not satisfy geometric constraints. However, their performance often degrades significantly in scenarios with a high outlier ratio. Consequently, deep learning-based outlier removal approaches have emerged as a mainstream alternative. As a pioneering effort, LFGC (Yi et al., 2018) introduces a permutation-invariant feature encoding mechanism to handle unordered correspondences and designs a task-specific loss function

to jointly optimize the prediction of inlier probability. Building on LFGC, OANet (Zhang et al., 2019) incorporates DiffPool and DiffUnpool layers to learn local context from sparse and unordered correspondences in a differentiable manner. PGFNet (Liu et al., 2023) proposes a Grouped Residual Attention mechanism to compute global preference scores and identify inliers. Li et al. (Li et al., 2023) design a U-shaped attention-based graph neural network, U-Match, which implicitly captures local contextual cues at multiple levels. T-Net++ (Xiao et al., 2024) introduces flexible "–" and "—" structures: the former iteratively learns correspondence relationships, while the latter integrates features to generate inlier weights. DeMatch (Zhang et al., 2024) decomposes motion fields to retain their dominant low-frequency components, enabling effective suppression of outliers and extraction of accurate motion vectors.

## 2.2 GNNs in Correspondences Learning

In recent years, GNNs have been widely adopted for two-view correspondence learning, owing to their strong capacity to capture consistency among correspondences. Specifically, NM-Net (Zhao et al., 2019) proposes a compatibility-specific mining method that discovers consistent neighbors through hierarchical extraction and aggregation of local correspondences. MS$^2$DGNet (Dai et al., 2022) constructs sparse semantic graphs at different levels and integrates multi-scale information to enrich contextual representation. CLNet (Zhao et al., 2021) prunes unreliable correspondences by computing both local and global consistency scores. NCMNet (Liu & Yang, 2023) explores graph-based contextual information across coordinate, feature, and global neighborhoods to identify reliable correspondences. GCT-Net (Guo et al., 2024) introduces a graph-guided context filtering mechanism to identify inliers and leverages their contextual consistency with other features to suppress outliers. BCLNet (Miao et al., 2024) learns local and global consensus in parallel using GNNs and explicitly models their interdependencies, enabling effective acquisition of bilateral consistency. However, previous works based on local neighborhood graphs fail to capture high-order geometric relationships among nodes effectively and lack the ability to model cross-subgraph structural consistency and dependencies. Thus, we design Energy-Guided Hypergraph Network (EGH-Net) to model intra-graph and inter-graph energy functions to address these limitations.

## 3 Methodoloy

### 3.1 Problem Formulation

Given a pair of matching images $[I, I']$, our goal is to remove outliers to obtain reliable correspondences and estimate the camera pose. We first extract $N$ keypoints from the images and then construct the initial correspondence set $S = \{s_i = (x_i, y_i, x_i', y_i') | i = 1, 2, ..., N\} \in \mathbb{R}^{N \times 4}$, where $(x_i, y_i)$ and $(x_i', y_i')$ denote the coordinates of the $i$-th keypoint in the source image $I$ and the target image $I'$, respectively.

However, due to the inherent ambiguity of feature descriptors, a large number of outliers are inevitably present in this initial correspondence set. To identify inliers and remove outliers, we propose the EGH-Net in Fig. 2, which consists of two Energy Function Guidance (EFG) modules, a Model Estimation, and a Full-Size Verification. Specifically, two EFG modules iteratively prune the initial correspondence set $S$ using the predicted logit values, producing two pruned sets $S_1 \in \mathbb{R}^{N_1 \times 4}$ and $S_2 \in \mathbb{R}^{N_2 \times 4}$. Then, the pruned set $S_2$ is fed to the Model Estimation module to compute the inlier weight $w$, which consists of a ResNet and an MLP layer. Subsequently, the weighted eight-point algorithm $g(\cdot)$ is applied to compute the essential matrix $E'$.

$$E' = g(S_2, w). \tag{1}$$

Finally, the estimated $E'$ is used for full-size verification on the initial correspondence set $S$ to recover inliers that are erroneously pruned.

$$y = h(E', S), \tag{2}$$

where $h(\cdot)$ computes the epipolar distances for full-size verification. $y$ is the epipolar distance set, which indicates the correctness of each correspondence in $S$.

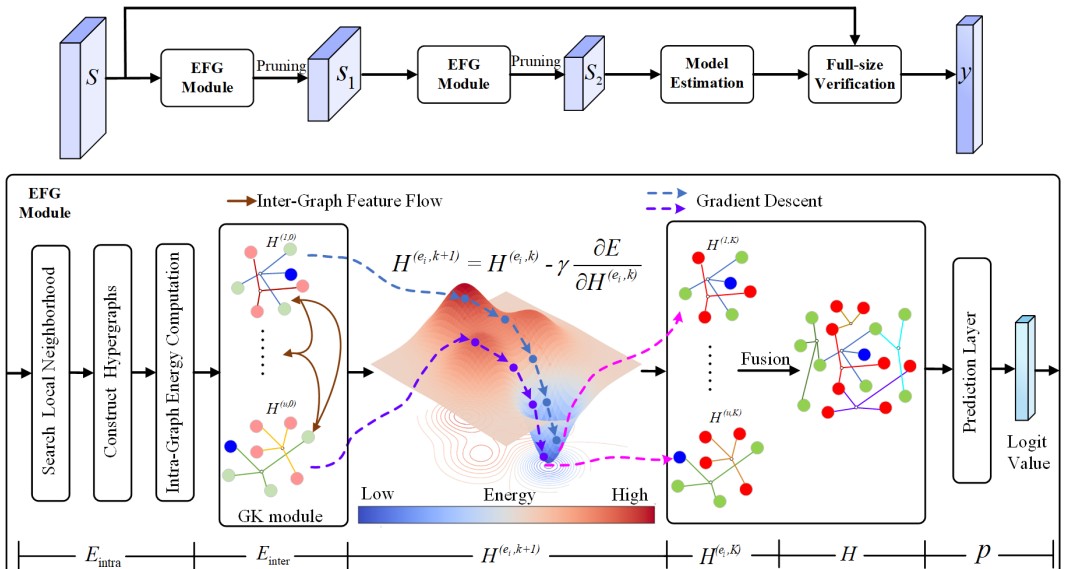

Figure 2: The architecture of the EGH-Net.

### 3.2 EFG MODULE

The EFG module is designed to update the feature representations and effectively model the high-order constraints of graph nodes based on two energy functions, as shown in Fig. 2. Specifically, the intra-graph energy $E_{\text{intra}}$ captures the consistency relationship within the subgraph by weighting the feature differences between nodes. The inter-graph energy $E_{\text{inter}}$ utilizes the Graph Kernel (GK) module to effectively capture structural correspondences across subgraphs, thereby achieving global structural alignment. Based on the above two energy terms, the EFG module dynamically updates the node features through the gradient descent and further fuses these updated representations $H^{(e_i, K)}$ into a global graph $H$. Thus, the EFG module can improve the accuracy and robustness of feature learning. The detailed introduction of each component in the EFG module is as follows.

***Intra-Graph Energy.*** To capture the consistency relationship between correspondences within subgraphs, we design an intra-graph energy function. Specifically, we first encode each correspondence $s_i$ into a feature map using Multi-Layer Perceptrons (MLPs). Subsequently, inspired by the graph construction methods in (Feng et al., 2019; Zhao et al., 2021; Feng et al., 2024; Chen et al., 2024), we search local neighborhoods to construct a hypergraph $\mathcal{G} = \{\mathcal{V}, \varepsilon\}$, where $\varepsilon = \{e_1, \ldots, e_u\}$ denotes the set of hyperedges $e_i$, and $u$ refers to the total number of hyperedges. $\mathcal{V} = \{V_{e_1}, ..., V_{e_n}\}$ represents the vertice set, and each $V_{e_i}$ is defined as $V_{e_i} = \{v_1, v_2, v_3, \ldots, v_{ne_i}\}$, where $ne_i$ refers to the number of vertices contained in hyperedge $e_i$. Then, we define intra-graph energy term $E_{\text{intra}}^{(e_i)}$ as the weighted feature differences between all nodes within $e_i$.

$$E_{\text{intra}}^{(e_i)} = \sum_{v_j, v_l \in V_{e_i}} ||H_j^{(e_i)} - H_l^{(e_i)}||_2^2, \tag{3}$$

where $H_j^{(e_i)} \in \mathbb{R}^{C \times D}$ and $H_l^{(e_i)} \in \mathbb{R}^{C \times D}$ represent the feature of the nodes $v_j$ and $v_l$ in hyperedge $e_i$, $C$ and $D$ represent the channel dimension and the feature dimension, respectively. To perform dynamic updates on the node representations, we compute the gradient of the energy term $E_{\text{intra}}^{(e_i)}$ concerning the node features $H^{(e_i)}$.

**Theorem 1 (Proven in Appendix A.2.)** Let $H^{(e_i)} \in \mathbb{R}^{C \times D \times ne_i}$ denote the feature matrix of the nodes in the hyperedge $e_i$, where $ne_i$ is the number of nodes of hyperedge $e_i$. Then, the gradient of the intra-energy $E_{\text{intra}}^{(e_i)}$ with respect to $H^{(e_i)}$ can be represented as:

$$\frac{\partial E_{\text{intra}}^{(e_i)}}{H^{(e_i)}} = 4 \cdot (ne_i \cdot I - 11^T) H^{(e_i)}, \tag{4}$$

where $I$ is the identity matrix, 1 is a matrix of ones.

***Inter-Graph Energy.*** Given two hyperedges $e_i$ and $e_m$ $(i \neq m)$ , we design the inter-graph energy $E_{\text{inter}}^{(e_i)}$ to capture the structural correspondences between different subgraphs.

$$E_{\text{inter}}^{(e_i)} = \sum_{m=1}^{u} \psi(||H^{(e_i)} - H^{(e_m)}||_F^2), \tag{5}$$

where $H^{(e_i)} \in \mathbb{R}^{C \times D \times ne_i}$ and $H^{(e_m)} \in \mathbb{R}^{C \times D \times ne_m}$ represent the feature matrices of hyperedges $e_i$ and $e_m$, respectively, with $ne_i$ and $ne_m$ indicating the number of nodes in each hyperedge. The function $\psi(\cdot)$ measures the difference between $e_i$ and $e_m$ based on the Frobenius norm $|| \cdot ||_F^2$. To progressively align node features with their subgraph structures, we compute the gradient of $H^{(e_i)}$.

$$\frac{\partial E_{\text{inter}}^{(e_i)}}{\partial H^{(e_i)}} = 2 \sum_{m=1}^{u} \psi'(||H^{(e_i)} - H^{(e_m)}||_F^2)(H^{(e_i)} - H^{(e_m)}), \tag{6}$$

where $\psi'(\cdot) = \frac{\partial \psi(\cdot)}{\partial H^{(e_i)}}$. However, considering $\psi(\cdot)$ depends on complex and potentially non-linear relationships between node features within and across subgraphs, it is often difficult to define explicitly. Existing approximation methods, such as fixed dot-product attention or kernel-based similarity functions (e.g., softmax-scaled inner products), suffer from two limitations: (1) their similarity computation is typically fixed and lacks adaptability to varying subgraph structures or scales, and (2) they exhibit quadratic computational complexity $O((ne_i)^2)$ ($ne_i$ is the number of nodes), making them inefficient for large and densely connected subgraphs. To this end, we propose the Graph Kernel (GK) module to approximate $\psi'(||H^{(e_i)} - H^{(e_m)}||_F^2)$, as shown in Fig. 3. Specifically, we first use MLPs and MaxPooling layers to sample the input subgraphs $H^{(e_i)}$ and $H^{(e_m)}$ into $h_t^{(e_i)}$ and $h_t^{(e_m)}$, with $T$ scales along the channel dimension. The operation is defined as follows:

$$h_t^{(e_i)} = \text{MaxPool}(\text{MLPs}(H^{(e_i)})), h_t^{(e_m)} = \text{MaxPool}(\text{MLPs}(H^{(e_m)})), \tag{7}$$

where $h_t^{(e_i)} \in \mathbb{R}^{C_t \times D \times ne_i}$, $h_t^{(e_m)} \in \mathbb{R}^{C_t \times D \times ne_m}$, and $t \in \{1, 2, \ldots, T\}$. The channel dimension of the $t$-th layer is $C_t = C/2^{t-1}$. Next, we extract global structural features through a top-down path using convolutions $Conv_1(\cdot)$ with a $1 \times 1$ kernel that performs feature transformation from the channel dimension $C_{t-1}$ to $C_t$.

$$h_t^{(e_i, up)} = h_t^{(e_i)} + Conv_1(h_{t-1}^{(e_i)}), \quad t \in \{2, 3, ..., T\}, \tag{8}$$

where $h_1^{(e_i, up)} = h_1^{(e_i)}$. We also design the bottom-up path that enhances local structural details through reverse convolutions $Conv_2(\cdot)$ with a $1 \times 1$ kernel. It performs feature transformation from the channel dimension $C_{t-1}$ to $C_t$.

$$h_t^{(e_m, down)} = h_t^{(e_m)} + Conv_2(h_{t+1}^{(e_m)}), \quad t \in \{T-1, T-2, ..., 1\}, \tag{9}$$

where $h_T^{(e_m, down)} = h_T^{(e_m)}$. We concatenate the features from the bidirectional paths at each scale and dynamically fuse them using point-wise convolutions $Conv_3(\cdot)$ with a $1 \times 1$ kernel. This produces $\hat{H}_{\text{inter}}^{(e_i, e_m)}$ to approximate the derivative $\psi'(||H^{(e_i)} - H^{(e_m)}||_F^2)$.

$$\hat{H}_{\text{inter}}^{(e_i, e_m)} = Conv_3(Cat(\{h_t^{(e_i, up)} + h_t^{(e_m, down)}\}_{t=1}^{T})). \tag{10}$$

Therefore, the gradient of the inter-graph energy $E_{\text{inter}}^{(e_i)}$ with respect to $H^{(e_i)}$ can be expressed as follow.

$$\frac{\partial E_{\text{inter}}^{(e_i)}}{\partial H^{(e_i)}} = 2 \sum_{m=1}^{u} \hat{H}_{\text{inter}}^{(e_i, e_m)} \left( H^{(e_i)} - H^{(e_m)} \right). \tag{11}$$

Thus, unlike traditional attention mechanisms, which focus solely on pairwise feature similarity, the GK module takes entire subgraphs $H^{(e_i)}$ and $H^{(e_m)}$ as input, modeling their global structure through a learnable parameter function, thus capturing structural similarity between subgraphs more effectively. Notably, the complexity of the GK module is dominated by concatenation and subsequent convolution, which has a complexity of $O(T \times ne_i)$.

***Gradient Descent.*** We combine the intra-graph and inter-graph energies to obtain $E = E_{\text{intra}} + E_{\text{inter}}$, then compute its gradient with respect to $H^{(e_i)}$ to update the hyperedge features. The total gradient of the hyperedge $e_i$ is obtained by combining the terms in Eq. 4 and Eq. 11.

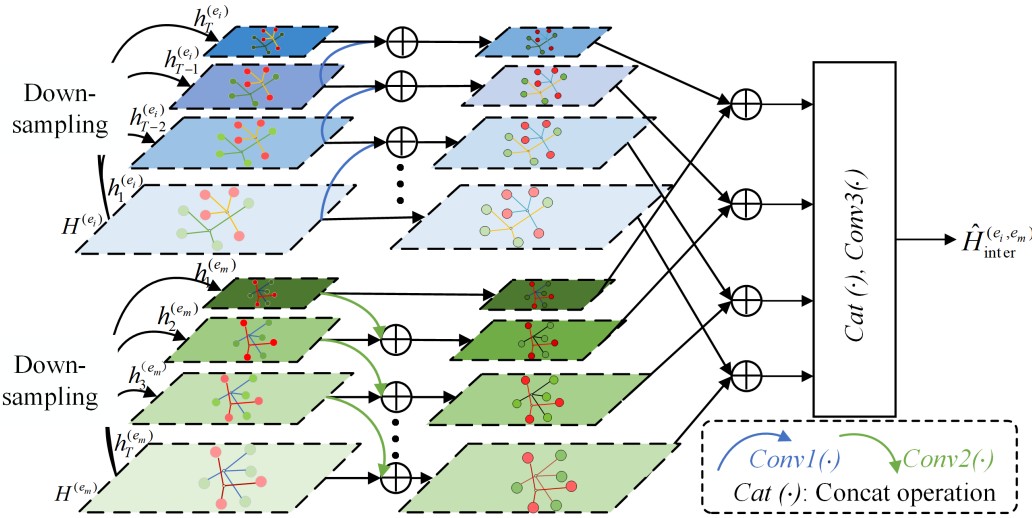

Figure 3: The architecture of the GK module.

$$\frac{\partial E}{\partial H^{(e_i)}} = \frac{\partial E_{\text{intra}}^{(e_i)}}{\partial H^{(e_i)}} + \frac{\partial E_{\text{inter}}^{(e_i)}}{\partial H^{(e_i)}} = 4 \cdot (ne_i \cdot I - 11^T)H^{(e_i)} + 2\sum_{m=1}^{u} \hat{H}_{\text{inter}}^{(e_i,e_m)}\left(H^{(e_i)} - H^{(e_m)}\right). \quad (12)$$

Based on Eq. 12, we use gradient descent to iteratively update the features of each hyperedge until convergence at the $k$-th iteration, resulting in $H^{(e_i,k)}$.

$$H^{(e_i,k)} = H^{(e_i,k-1)} - \gamma \cdot \frac{\partial E}{\partial H^{(e_i,k-1)}}$$

$$= H^{(e_i,k-1)} - 4\gamma \cdot (ne_i \cdot I - 11^T)H^{(e_i,k-1)} - 2\gamma \cdot \sum_{m=1}^{u} \hat{H}_{\text{inter}}^{(e_i,e_m,k-1)}\left(H^{(e_i,k-1)} - H^{(e_m,k-1)}\right). \quad (13)$$

where $\gamma$ denotes the learning rate of gradient descent. $H^{(e_i,0)} = H^{(e_i)}$, and $H^{(e_i,k)} \in \mathbb{R}^{C \times ne_i \times D}$ denotes the features at the $k$-th iteration. In this way, the model iteratively adjusts the node features to better satisfy higher-order constraints. Subsequently, we fuse the representations from all graphs to obtain the final representation $H$ used to estimate inlier probabilities.

$$H = F(H^{(e_1,1)}, ..., H^{(e_u,K)}), \quad (14)$$

where $F(\cdot)$ is the fusion function, including a concatenation operation and a ResNet block. Finally, we use MLPs, tanh, and ReLU activations to map the fused representation $H$ to the predicted probabilities $p$ for inlier/outlier classification.

### 3.3 LOSS FUNCTION

To optimize EGH-Net, we employ a hybrid loss function comprising two components: a matrix regression loss $L_e$ and a classification cross-entropy loss $L_c$. The matrix regression loss $L_e$ is designed to evaluate the geometric discrepancy between the ground-truth essential matrix $E$ and the model's predicted estimate $E'$, and $\alpha$ is a weight parameter.

$$L = L_c + \alpha L_e(E, E'). \quad (15)$$

## 4 EXPERIMENTS

### 4.1 IMPLEMENTATION DETAILS

**Baseline methods.** We compare the EGH-Net with several representative approaches, including a traditional method RANSAC (Fischler & Bolles, 1987), and nine learning-based networks LFGC (Yi et al., 2018), OANet (Zhang et al., 2019), CLNet (Zhao et al., 2021), MS²DGNet (Dai et al.,

Table 1: The evaluation on the YFCC100M dataset with different descriptors for camera pose estimation.

| Matchers | Params (MB) | SIFT | | | | SuperPoint | | | |
|---|---|---|---|---|---|---|---|---|---|
| | | Known(%) | | Unknown(%) | | Known(%) | | Unknown(%) | |
| | | 5° | 20° | 5° | 20° | 5° | 20° | 5° | 20° |
| RANSAC | - | 5.81 | 16.88 | 9.07 | 22.92 | 12.85 | 31.22 | 17.47 | 38.83 |
| LFGC | **0.39** | 13.81 | 35.20 | 23.95 | 52.44 | 12.18 | 34.75 | 24.25 | 52.70 |
| OANet | 2.47 | 32.57 | 56.89 | 38.95 | 66.85 | 29.52 | 53.76 | 35.27 | 66.81 |
| CLNet | 1.27 | 39.00 | 62.48 | 54.05 | 75.76 | 27.56 | 50.82 | 39.19 | 67.37 |
| MS$^2$DGNet | 2.61 | 38.36 | 76.04 | 49.13 | 76.04 | 31.15 | 55.16 | 39.19 | 70.36 |
| U-Match | 7.76 | 46.78 | 68.75 | 60.22 | 80.26 | 35.01 | 56.80 | 44.29 | 70.90 |
| NCMNet | 4.77 | 52.39 | 72.54 | 63.52 | 82.54 | 38.92 | 61.28 | 48.20 | 74.71 |
| T-Net++ | 4.11 | 42.69 | 65.85 | 51.15 | 75.37 | 37.24 | 59.69 | 42.80 | 72.80 |
| DeMatch | 5.86 | 49.37 | 70.55 | 61.82 | 81.02 | 39.73 | 60.97 | 48.55 | 74.36 |
| BCLNet | 4.50 | 54.67 | 74.04 | 66.38 | 83.80 | 40.99 | 63.51 | 50.30 | 76.56 |
| EGH-Net | 2.81 | **59.43** | **78.01** | **71.05** | **86.47** | **44.68** | **67.23** | **52.87** | **78.25** |

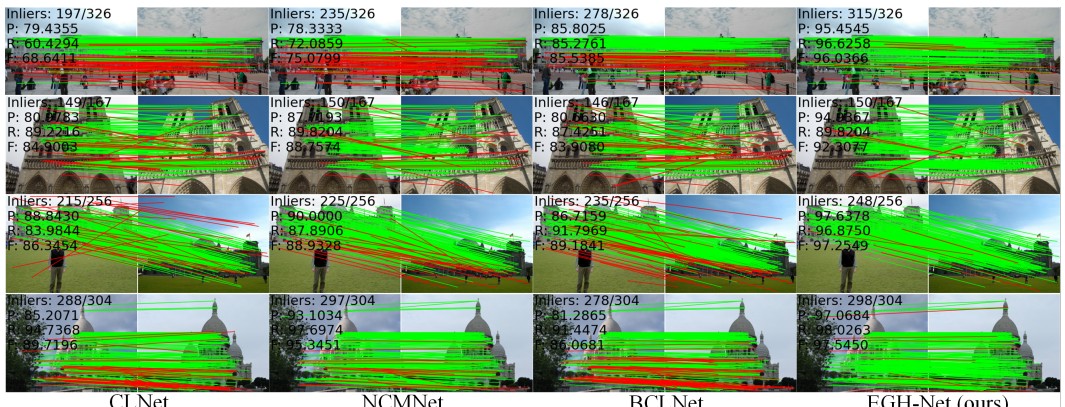

Figure 4: Visualization results of different methods. The green lines and red lines represent inliers and outliers, respectively.

2022), U-Match (Sun et al. 2020), NCMNet (Liu & Yang, 2023), T-Net++ (Xiao et al., 2024), DeMatch (Zhang et al., 2024), and BCLNet (Miao et al., 2024).

**Training details.** The input of EGH-Net is an $N \times 4$ initial correspondences, extracted using SIFT or SuperPoint. We adopt the same hypergraph construction strategy as (Chen et al., 2024). For computational simplicity and structural consistency, the number of nodes in each hypergraph is fixed to 18 in the EFG module. The channel dimension is set to 128. $\gamma$ denotes the gradient descent learning rate, learned as a parameter and initialized to 1. We set $T = 4$ in the GK module. (Please see Appendix A.5 for the ablation study of $T$). We use the Adam optimizer (Kingma & Ba, 2014) to train the EGH-Net for 500000 iterations. We also use a warm-up strategy (Luanyuan et al., 2024; Yang et al., 2024) to adjust the learning rate, where it increases linearly from 0 to $10^{-3}$ in the first 10000 iterations, after that the learning rate begins to decrease and is reduced for every 20000 iterations with a factor of 0.4. All experiments are conducted on the Ubuntu 18.04 operating system with an NVIDIA RTX 4090 GPU. The weight parameter $\alpha$ of the loss in Eq. (15) is set to 0.5.

## 4.2 Camera Pose Estimation

The goal of camera pose estimation is to utilize identified inliers to accurately compute the relative rotation and translation matrix between different camera views. Here, we first train EGH-Net on the Yahoo's YFCC100M dataset in known and unknown scenes (Thomee et al., 2016) using both SIFT (Fischler & Bolles, 1987) and SuperPoint(DeTone et al., 2018) descriptors to estimate camera poses. For performance assessment, we adopt mean Average Precision (mAP) at angular error thresholds of 5° and 20° (i.e., mAP5° and mAP20°). We compare EGH-Net with baselines as shown in Ta-

ble 1. EGH-Net achieves a 4.76% and 4.67% performance improvement over BCLNet on known and unknown outdoor scenes with SIFT descriptor, respectively, but with 37.56% fewer training parameters. Besides, using the SuperPoint descriptor, EGH-Net also achieves leading performance, outperforming BCLNet by 3.69% and 2.57% on known and unknown scenes, respectively.

Fig. 4 visualizes the predicted correspondences, where we employ the inlier ratio, Precision (P), Recall (R), and F-score (F) (reported in the upper left corner of each subfigure) to quantitatively evaluate the matching quality of each image pair. We observe that EGH-Net is capable of eliminating more outliers than CLNet, NCMNet, and BCLNet, thereby significantly improving matching accuracy. Furthermore, we evaluate the effectiveness of EGH-Net on the more complex indoor SUN3D dataset (Xiao et al., 2013) for camera pose estimation, under both with/without RANSAC settings for known and unknown scenes. As reported in Table 2, our EGH-Net outperforms all baseline methods, achieving the best performance across mAP5° under various challenging scenes. Ad-

Table 2: Evaluation on SUN3D dataset with SIFT for camera pose estimation. mAP5°(%) is reported.

| Matchers | Known (%) | | Unknown (%) | |
|---|---|---|---|---|
| | - | RANSAC | - | RANSAC |
| RANSAC | - | 19.13 | - | 14.57 |
| LFGC | 11.55 | 20.60 | 9.30 | 16.40 |
| OANet | 20.86 | 22.31 | 16.18 | 17.18 |
| CLNet | 20.35 | 24.15 | 17.03 | 18.52 |
| MS$^2$DGNet | 22.20 | 23.00 | 17.84 | 17.79 |
| U-Match | 24.99 | 23.32 | 21.38 | 18.13 |
| NCMNet | 25.80 | 25.72 | 20.82 | 20.33 |
| T-Net++ | 23.48 | 23.76 | 18.62 | 17.70 |
| DeMatch | 29.53 | 24.02 | 23.92 | 19.24 |
| BCLNet | 24.35 | 25.27 | 20.14 | 20.00 |
| EGH-Net | **30.08** | **26.75** | **24.60** | **21.45** |

ditionally, we conduct the outlier rejection in the Appendix A.4, where Table 8 shows that the EGH-Net outperforms all baselines in terms of the F-score. Please see Appendix A.4 for details.

Besides, we compare camera pose estimation performance (mAP5°) and computational complexity across different methods in Fig. 5. EGH-Net achieves superior accuracy for camera pose estimation while maintaining fewer parameters and shorter inference time. Please see Appendix A.6 for details.

### 4.3 HOMOGRAPHY ESTIMATION

The purpose of homography estimation is to determine the transformation matrix that relates two views of the same scene. Here, we conduct experiments on the HPatches dataset (Balntas et al., 2017) for homography estimation, which contains 116 diverse scenes, and each consists of 6 images with corresponding homography references. We first use the SIFT detector to extract keypoints and compute descriptors, to form the initial correspondences.

Table 3: Evaluation on HPatches for homography estimation.

| Matchers | ACC.@3PX | ACC.@5PX | ACC.@10PX |
|---|---|---|---|
| LFGC | 67.93 | 82.59 | **92.76** |
| OANet | 69.66 | 82.93 | 91.90 |
| CLNet | 69.83 | 81.55 | 90.69 |
| MS$^2$DGNet | 68.28 | 79.66 | 90.50 |
| NCMNet | 70.69 | 81.90 | 91.03 |
| EGH-Net | **71.90** | **83.45** | **92.76** |

Then, we train the EGH-Net on the YFCC100M dataset to remove outliers between the image pairs. We further calculate the percentage of image pairs with mean reprojection errors smaller than 3, 5, and 10 pixels and compare our method with existing approaches. Table 3 shows that EGH-Net outperforms baseline methods in all error thresholds, especially under the stricter 3-pixel threshold.

### 4.4 VISUAL LOCALIZATION

The visual localization task aims to infer the 6-DOF camera pose by identifying reliable correspondences between the query image and the reference image. We use the Aachen Day-Night dataset (Sattler et al., 2018) for visual localization experiments, which includes 4328 reference images and 922 query images (including 824 daytime and 98 nighttime images). Specifically, we first utilize the SIFT detector to extract 4096 initial correspondences for each image in the Aachen Day-Night dataset. Then, we train the EGH-Net on the YFCC100M dataset

Table 4: Evaluation of visual localization on the Aachen Day-Night dataset.

| Matchers | Day (%) | | | Night (%) | | |
|---|---|---|---|---|---|---|
| | (0.25m, 2°) | (0.5m, 5°) | (1.0m, 10°) | | | |
| LFGC | 81.3 | 91.4 | 95.9 | 68.4 | 78.6 | 87.8 |
| OANet | 82.3 | 91.9 | 96.5 | 71.4 | 79.6 | 90.8 |
| CLNet | 83.3 | 92.4 | 97.0 | 71.4 | 80.6 | **93.9** |
| MS$^2$DGNet | 82.8 | 92.1 | 96.8 | 70.4 | 82.7 | **93.9** |
| NCMNet | 83.1 | 91.4 | 96.8 | 69.4 | 80.7 | 89.8 |
| BCLNet | 83.0 | 91.6 | **97.5** | 68.4 | 79.6 | 87.8 |
| EGH-Net | **85.1** | **92.7** | 97.2 | **73.5** | **85.7** | 93.9 |

to remove outliers and obtain reliable corre-
spondences. Next, we match nighttime images with daytime images of known poses, combining
model results with COLMAP's correspondences. For localization evaluation, we use HLoc (Sarlin
et al., 2019) metrics to calculate the success rate of image registration under three threshold settings
in terms of distance and orientation thresholds. Table 4 shows that EGH-Net performs well, partic-
ularly under the most challenging threshold (0.25m, 2°) in both daytime and nighttime conditions.

### 4.5 ABLATION STUDIES

**Intra and inter-graph ablations.** To evaluate
the impact of the intra-graph energy term $E_{\text{intra}}$
and the inter-graph energy term $E_{\text{inter}}$ on the
camera pose estimation task, we train four ver-
sions of EGH-Net, including a full EGH-Net,
a version without $E_{\text{intra}}$, one without $E_{\text{inter}}$, and
one without both. As shown in Table 5, remov-
ing either $E_{\text{intra}}$ or $E_{\text{inter}}$ leads to a clear drop
in performance, while the complete EGH-Net
achieves the best results. This suggests that

Table 5: Ablation study on intra-graph and inter-graph energy terms on YFCC100M with unknown scenes.

| Method | mAP5° | mAP20° |
|---|---|---|
| EGH-Net | 70.65 | 86.54 |
| EGH-Net w/o $E_{\text{intra}}$ | 68.95 | 85.26 |
| EGH-Net w/o $E_{\text{inter}}$ | 68.88 | 85.12 |
| EGH-Net w/o $E_{\text{intra}}$ & w/o $E_{\text{inter}}$ | 67.05 | 84.43 |

jointly leveraging both intra- and inter-graph energy terms facilitates more effective node feature
optimization and improves the model's ability to reject outliers.

**Hypergraph vs. traditional graph.** To assess the effect of hypergraphs on camera pose estimation,
we replace the hypergraph structure in EGH-Net with a traditional graph (Zhao et al., 2021) for
modeling consistency between correspondences. Then, we compare the performance of graph-based
EGH-Net and hypergraph-based EGH-Net for the camera pose estimation task in Table 6. The
hypergraph-based EGH-Net achieves improvements of 1.43% over the traditional graph model in
mAP5°. This comparison that hypergraphs capture higher-order constraints more effectively than
traditional graphs, thereby improving the accuracy of outlier removal.

**GK module vs. transformer.** To examine the effectiveness of the GK module, we replace the
GK module with a standard transformer (Vaswani et al., 2017) and check whether the performance
drops. As reported in Table 7, replacing the GK module with a transformer results in a 1.41%
drop in mAP5° and a parameter increase of approximately 14.95%. These results demonstrate the
effectiveness of the GK module in capturing inter-graph relationships, and achieving better matching
accuracy while maintaining superior parameter efficiency.

Table 6: Graph and Hypergraph comparison on YFCC100M with unknown scenes.

| Method | mAP5° | mAP10° | mAP15° | mAP20° |
|---|---|---|---|---|
| Graph | 69.22 | 77.60 | 82.47 | 85.43 |
| Hypergraph | 70.65 | 79.31 | 83.82 | 86.54 |

Table 7: Comparison of efficiency and accuracy between Transformer and GK module.

| Method | mAP5° | mAP20° | Params (MB) |
|---|---|---|---|
| Transformer | 69.24 | 85.71 | 3.23 |
| GK module | 70.65 | 86.54 | 2.81 |

## 5 CONCLUSION AND DISCUSSION

In this paper, we propose the EGH-Net, an energy-guided hypergraph network, which effectively
models higher-order constraints and suppresses outliers for robust correspondence learning. Specif-
ically, to enable the hypergraph to accurately capture higher-order constraints among nodes, we
first introduce the intra-graph energy function, which models the consistency within each subgraph
by evaluating the differences among node features. Then, we design the GK module to compute
inter-graph energy for aligning structures between subgraphs. Finally, by integrating intra-graph
and inter-graph energy terms, EGH-Net dynamically updates node features via gradient descent,
effectively modeling high-order constraints and eliminating outliers. Experiments on different tasks
and datasets demonstrate that the performance of EGH-Net surpasses other state-of-the-art methods.

ETHICS STATEMENT

This paper proposes EGH-Net, an energy-guided hypergraph network for two-view correspondence learning. By constructing intra- and inter-graph energy functions to model higher-order consistency constraints, the method effectively suppresses outliers and enhances inlier relationships, providing new insights into robust feature matching. This study involves no ethical concerns.

REPRODUCIBILITY STATEMENT

To ensure reproducibility, we provide details of the proposed method, training setup, and hyperparameter settings in Section 4.1, and theoretical proofs in Section A.2. All datasets used are publicly available and described in Section A.3. Furthermore, we will release the code when the paper is accepted.

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

## A APPENDIX

### A.1 STATEMENT ON THE USE OF LARGE LANGUAGE MODELS

This paper only employed large language models (LLMs) for linguistic polishing of the manuscript. No parts of the methodology, experiments, or analysis were generated or influenced by LLMs.

### A.2 THE DERIVATION DETAILS OF THEOREM 1.

To capture the consistency relationship between correspondences within subgraphs, we design an intra-graph energy function. Specifically, we first encode each correspondence $s_i$ into a feature map using Multi-Layer Perceptrons (MLPs). Subsequently, inspired by the graph construction methods in (Feng et al., 2019; Zhao et al., 2021; Feng et al., 2024; Chen et al., 2024), we search local neighborhoods to construct a hypergraph $\mathcal{G} = \{\mathcal{V}, \varepsilon\}$, where $\varepsilon = \{e_1, \ldots, e_u\}$ denotes the set of hyperedges $e_i$, and $u$ refers to the total number of hyperedges. $\mathcal{V} = \{V_{e_1}, ..., V_{e_n}\}$ represents the vertice set, and each $V_{e_i}$ is defined as $V_{e_i} = \{v_1, v_2, v_3, \ldots, v_{ne_i}\}$, where $ne_i$ refers to the number of vertices contained in hyperedge $e_i$. Then, we define intra-graph energy term $E_{\text{intra}}^{(e_i)}$ as the weighted feature differences between all nodes within $e_i$.

$$E_{\text{intra}}^{(e_i)} = \sum_{v_j, v_l \in V_{e_i}} ||H_j^{(e_i)} - H_l^{(e_i)}||_2^2, \qquad (16)$$

where $H_j^{(e_i)} \in \mathbb{R}^{C \times D}$ and $H_l^{(e_i)} \in \mathbb{R}^{C \times D}$ represent the feature of the nodes $v_j$ and $v_l$ in hyperedge $e_i$, $C$ and $D$ represent the channel dimension and the feature dimension, respectively. To perform dynamic updates on the node representations, we compute the gradient of the energy term $E_{\text{intra}}^{(e_i)}$ concerning the node features $H^{(e_i)}$. Expanding the squared Euclidean distance.

$$\| H_j^{(e_i)} - H_l^{(e_i)} \|_2^2 = \| H_j^{(e_i)} \|_2^2 + \| H_l^{(e_i)} \|_2^2 - 2(H_j^{(e_i)})^T H_l^{(e_i)}. \qquad (17)$$

Substituting this into Eq. (16), we obtain:

$$\begin{aligned} E_{\text{intra}}^{(e_i)} &= \sum_{v_j, v_l \in V_{e_i}} \left( \| H_j^{(e_i)} \|_2^2 + \| H_l^{(e_i)} \|_2^2 - 2(H_j^{(e_i)})^T H_l^{(e_i)} \right) \\ &= 2ne_i \sum_{j \in V_{e_i}} \| H_j^{(e_i)} \|_2^2 - 2 \sum_{v_j, v_l \in V_{e_i}} (H_j^{(e_i)})^T H_l^{(e_i)}. \end{aligned} \qquad (18)$$

Let $H^{(e_i)} = \{H_j^{(e_i)}\}_{j=1}^{n_{e_i}}$, where each node feature is represented as $H_j^{(e_i)} \in \mathbb{R}^{C \times D}$. By reshaping all node features into a matrix $H^{(e_i)} \in \mathbb{R}^{(C \cdot D) \times n_{e_i}}$, the intra-graph energy can be expressed in trace form as:

$$\begin{aligned} E_{\text{intra}} &= 2n_{e_i} \cdot \text{Tr}(H^{(e_i)T} H^{(e_i)}) - 2 \cdot \text{Tr}(H^{(e_i)T} 11^T H^{(e_i)}) \\ &= 2 \cdot \text{Tr}\left( H^{(e_i)T} \left( n_{e_i} \cdot I - 11^T \right) H^{(e_i)} \right), \end{aligned} \qquad (19)$$

where $I$ is the identity matrix, $1 \in \mathbb{R}^{C \times ne_i \times ne_i}$ is a matrix of ones. $ne_i \cdot I - 11^T$ is a smoothing constraint matrix designed to capture the consistency relationship among nodes within hyperedge $e_i$.

To perform dynamic updates on the node representations, we compute the gradient of the energy term $E_{\text{intra}}^{(e_i)}$ concerning the node features $H^{(e_i)}$.

$$\frac{\partial E_{\text{intra}}^{(e_i)}}{H^{(e_i)}} = 4 \cdot (ne_i \cdot I - 11^T) H^{(e_i)}. \qquad (20)$$

By minimizing the energy of node feature representations within each hyperedge, we ensure that geometrically related nodes exhibit consistency in their feature representations.

### A.3 DATASET

**YFCC100M Dataset and SUN3D Dataset.** To evaluate the robustness of the proposed method in different environments, we selected two representative datasets: YFCC100M Thomee et al. (2016)

and SUN3D Xiao et al. (2013), corresponding to large-scale outdoor and indoor scenes, respectively. YFCC100M is a large-scale multimedia dataset containing 100 million publicly licensed images and videos from Flickr, which is widely used in tasks such as image matching, localization, and 3D reconstruction. According to the baseline (Zhang et al., 2019), we selected 68 image sequences for training, covering a variety of outdoor conditions and viewpoint changes. The SUN3D dataset provides densely collected indoor RGB-D image sequences with rich layout, occlusion, and structural changes, which are an ideal complement for evaluating the robustness of the model in complex indoor geometric scenes. The dataset contains a total of 254 sequences, of which 239 are used for training and the rest are used for testing. The training part is further divided into training set (60%), validation set (20%), and test set (20%). The unknown sequences are the test sequences described above. We use mAP5° and mAP20° as the evaluation indicators of camera pose estimation performance.

**HPatches Dataset.** HPatches dataset Balntas et al. (2017) serves as a standard benchmark dataset for evaluating the performance of local feature descriptors, and is widely adopted in tasks such as feature matching, image registration, and patch retrieval. It consists of 116 image sequences, each containing one reference image and five additional images that exhibit variations in either viewpoint or illumination, thereby simulating a range of real-world perturbations. This dataset enables a systematic assessment of the robustness of descriptors under diverse scene changes and geometric or photometric transformations.

**Aachen Day-Night Dataset** The Aachen Day-Night dataset Sattler et al. (2018) is a challenging benchmark designed for evaluating image-based localization methods under extreme illumination changes, particularly between daytime and nighttime conditions. It consists of high-resolution images captured in the old town of Aachen, Germany, including a set of daytime reference images with accurate ground-truth poses and a set of nighttime query images that require localization within the same 3D scene. This dataset is especially suitable for downstream tasks such as visual localization based on image feature matching, as it provides a realistic setting to test the robustness of local features and matching algorithms under severe lighting variations.

## A.4 OUTLIER REJECTION.

Outlier rejection serves as a critical component in many computer vision tasks, such as visual localization, camera pose estimation, and image retrieval. In this section, we conduct a detailed performance comparison among several outstanding outlier rejection methods. Table 8 shows the experimental results on both YFCC100M and SUN3D datasets with unknown scenes, evaluated using three standard metrics: Precision, Recall, and F-score. EGH-Net consistently achieves the best results of Precision and F-score across the two datasets. Notably, EGH-Net effectively rejects many outliers through a pruning strategy, thereby enhancing the reliability of the matching process. However, this strategy also discards some true correspondences that should have been preserved, leading to missed detections and a reduction in Recall. Despite the decrease in Recall, the removal of outliers significantly improves Precision, resulting in an optimal F-score.

## A.5 IMPACTS OF THE NUMBER OF SCALES T IN THE GK MODULE.

To assess the impact of the number of scales $T$ in the GK module, we conduct an ablation study by varying $T$. Table 9 presents the performance on the known and unknown scenes. As $T$ increases, performance generally improves, reaching its peak at $T = 4$. These results indicate that introducing deeper multi-scale interactions facilitates more effective modeling of graph similarity. However, further increasing $T$ to 5 or 6 results in a slight performance drop, possibly due to excessive scale ranges during sampling. Therefore, we adopt $T = 4$ as the default setting to balance modeling capacity and computational efficiency.

## A.6 PERFORMANCE AND COMPUTATION COMPLEXITY ANALYSIS.

To evaluate model performance, we compare various methods in terms of mAP5°, parameters (Params), floating-point operations (FLOPs), and inference time (Time) for camera pose estimation. As shown in Fig. 5, the lightweight network CLNet requires the fewest Params, FLOPs, and Time, but it exhibits relatively poor performance in mAP5°. In contrast, BCLNet and NCMNet achieve

Table 8: Quantitative comparative results of outlier rejection experiment on both YFCC100M and SUN3D datasets. Bold represents the best

| Matcher | YFCC100M | | | SUN3D | | |
|---|---|---|---|---|---|---|
| | Precision (%) | Recall (%) | F-score(%) | Precision (%) | Recall (%) | F-score(%) |
| RANSAC | 41.83 | 57.08 | 48.28 | 44.11 | 46.42 | 45.24 |
| OANet++ | 55.78 | 85.93 | 67.65 | 46.15 | 84.36 | 59.66 |
| CLNet | 75.05 | 76.41 | 75.72 | 60.01 | 68.09 | 63.80 |
| MS2DGNet | 59.11 | 88.40 | 70.85 | 46.95 | 84.55 | 60.37 |
| NCMNet | 77.07 | 78.27 | 77.41 | 60.91 | 68.67 | 63.90 |
| T-Net++ | 59.16 | 86.35 | 70.21 | 47.81 | 84.27 | 61.01 |
| DeMatch | 60.74 | **90.30** | 72.66 | 48.32 | **84.99** | 61.62 |
| BCLNet | 77.72 | 79.31 | 78.53 | 60.45 | 68.45 | 64.20 |
| **EGH-Net** | **79.51** | 80.01 | **79.75** | **62.15** | 70.08 | **65.86** |

Table 9: Impact of different scales $T$ in the GK module on YFCC100M with unknown scenes.

| T | Known | | UnKnown | |
|---|---|---|---|---|
| | mAP5° | mAP20° | mAP5° | mAP20° |
| 1 | 57.56 | 76.85 | 68.92 | 85.40 |
| 2 | 58.19 | 77.06 | 69.37 | 85.81 |
| 3 | 58.37 | 77.41 | 70.01 | 86.18 |
| 4 | 59.15 | 77.90 | 70.65 | 86.54 |
| 5 | 59.03 | 77.23 | 70.38 | 86.06 |
| 6 | 58.57 | 76.92 | 69.83 | 85.77 |

superior performance at the cost of higher computational demands in terms of Params, FLOPs, and Time. EGH-Net stands out by achieving high mAP5° while maintaining lower Params and Time.

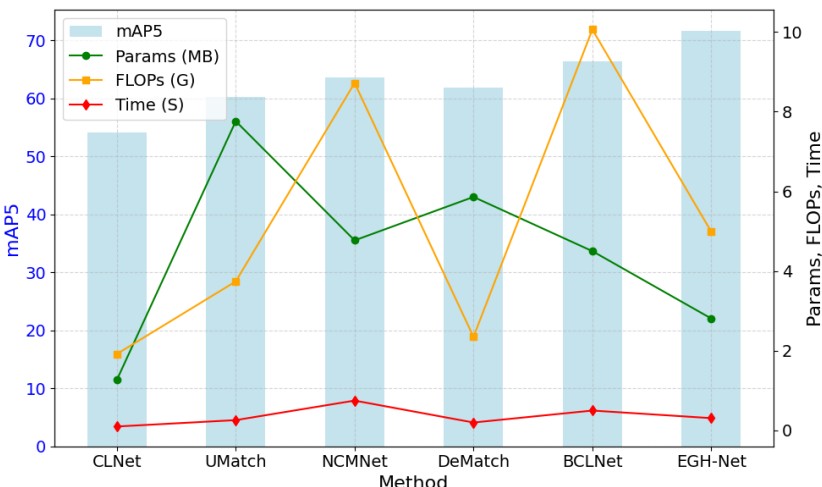

Figure 5: Visualization of performance comparison across different methods on YFCC100M dataset with SIFT for camera pose estimation.