# OpenReview forum: "EGH-Net: Energy-Guided Hypergraph for Two-View Correspondence Learning"
_ICLR.cc/2026/Conference — ICLR 2026 Conference Withdrawn Submission_

### Official Review · Reviewer_Z17h · 2025-10-28

**Soundness:** 3
**Presentation:** 3
**Contribution:** 2
**Rating:** 4
**Confidence:** 5

**Summary:**

This paper introduces a new method, EGH-Net, for correspondence pruning. The goal is to remove false matches and retain correct matches. The method replaces pairwise graphs with hypergraphs guided by energy functions to capture higher-order geometric consistency. The energy functions include an intra-graph energy term and an inter-graph energy term, capturing local information and global information, respectively. These energy terms jointly optimize node features via gradient descent, improving the distinction between inliers and outliers. Given the initial correspondences built using SIFT or SuperPoint, EGH-Net achieves state-of-the-art performance on multiple datasets, including YFCC100M, SUN3D, HPatches, and Aachen Day-Night.

**Strengths:**

This paper introduces an energy-guided hypergraph. Compared with the existing graph-based methods, the hypergraph effectively models higher-order geometric constraints beyond pairwise relations.

The use of intra and inter graph energy functions with a learnable Graph Kernel module optimize the node features through stochastic gradient descent, incorporating both local and global consistency.

The authors conduct experiments on multiple datasets, YFCC100M, SUN3D, HPatches, and Aachen Day-Night. EGH-Net achieves the SOTA performance in camera pose estimation, homography estimation, and visual localization.

**Weaknesses:**

With the development of feature matching, correspondence pruning is not as important as it used to be. Some advanced feature matching methods [1,2,3,4,5] are able to establish high-quality correspondences. A vanilla RANSAC can get a quite accurate model estimate based on these correspondences. However, the authors only conduct experiments using SIFT and SuperPoint, which are relatively outdated. The experiments do not clearly show the added value of the proposed approach in today’s feature matching landscape.

Some important details and analysis are missing. For instance, the authors use gradient descent to update node features, which may lead to some overhead. The number of iterations used for this optimization is not reported. The experiments regarding the efficiency, such as time consumption, are missing.

In Sec. 3.2, the authors use a Graph Kernel module to approximate the derivative. However, since this module is implemented using MLP and pooling layers, it is unclear how or why this design provides a valid approximation of the derivative term. A theoretical explanation would help clarify the soundness of this approximation.

[1] Sarlin, Paul-Edouard, et al. "Superglue: Learning feature matching with graph neural networks." *Proceedings of the IEEE/CVF conference on computer vision and pattern recognition*. 2020.

[2] Jiang, Hanwen, et al. "Omniglue: Generalizable feature matching with foundation model guidance." *Proceedings of the IEEE/CVF Conference on Computer Vision and Pattern Recognition*. 2024.

[3] Edstedt, Johan, et al. "Roma: Robust dense feature matching." *Proceedings of the IEEE/CVF Conference on Computer Vision and Pattern Recognition*. 2024.

[4] Leroy, Vincent, Yohann Cabon, and Jérôme Revaud. "Grounding image matching in 3d with mast3r." *European Conference on Computer Vision*. Cham: Springer Nature Switzerland, 2024.

[5] Xue, Fei, et al. "MATCHA: Towards matching anything." *Proceedings of the Computer Vision and Pattern Recognition Conference*. 2025.

**Questions:**

In Line 205-206, what do C and D stand for? Why is the node feature represented as a 2D matrix instead of a 1D vector?

---

### Official Review · Reviewer_U5vB · 2025-10-30

**Soundness:** 2
**Presentation:** 2
**Contribution:** 1
**Rating:** 2
**Confidence:** 5

**Summary:**

The paper tackles robust two-view correspondence learning, where local-graph GNNs struggle to encode higher-order geometric constraints and thus fail at outlier rejection in challenging scenes. It proposes EGH-Net, which builds a hypergraph over correspondences and optimizes node features by minimizing two energy terms: an intra-graph consistency energy and an inter-graph structural-similarity energy approximated by a learnable Graph Kernel module with multi-scale decomposition. Across pose estimation on YFCC100M and SUN3D, homography on HPatches, and localization on Aachen Day–Night, the method reports higher mAP and accuracy than recent baselines while using fewer parameters in several settings.

**Strengths:**

1. The problem formulation is concrete and aligns with known failure modes of KNN-based neighborhood graphs in correspondence pruning. By moving to hyperedges, the method can encode group-wise relations that are otherwise hard to capture with pairwise edges alone.

2. The experimental scope spans several standard downstream tasks with consistent data protocols, which helps readers gauge transferability. Showing both SIFT and SuperPoint descriptors adds a small but useful check on descriptor dependence.
3. The ablations are straightforward and interpretable: removing either intra- or inter-graph energy hurts, and replacing the hypergraph by a traditional graph or the GK by a transformer degrades accuracy and increases parameters. These results support the claim that both the representation and the energy design matter.

**Weaknesses:**

1. The novelty is limited relative to prior hypergraph correspondence models and energy-based consistency formulations. The intra-graph term reduces to a Laplacian-like smoothness penalty with an unsurprising gradient, and the inter-graph term is essentially a learned similarity on subgraph features; the paper does not convincingly isolate what is conceptually new beyond this combination. A controlled comparison against recent hypergraph or bilateral-consensus GNNs with matched capacity and training would be needed to justify an ICLR-level methodological advance.
2. The theoretical depth is thin where it matters most. The only stated theorem gives a trivial gradient form for the intra-graph energy, whereas the harder piece, the learnable approximation to the inter-graph energy derivative, is left without guarantees about stability, bias, or convergence of the alternating feature updates. Without bounding the approximation error of the GK module or showing monotonic energy decrease in practice, the optimization story remains heuristic.
3. The empirical gains, while present, are often modest and lack uncertainty quantification. For instance, mAP5° improvements over strong baselines are single-digit points, and the paper does not report confidence intervals, per-seed variance, or statistical tests across runs, which is important given dataset stochasticity and pruning sensitivity. The Aachen and HPatches wins are encouraging but would be more convincing with robust error bars and seed aggregation.
4. The complexity and scalability claims are not rigorously substantiated. Equation (11) sums inter-graph interactions over all hyperedges, yet the text attributes GK complexity mainly to concatenation and 1×1 convolutions with O(T×nei), which appears to ignore the u-wise summation term and practical memory traffic from multi-scale tensors. A FLOPs and wall-clock breakdown against transformer baselines on the same hardware and batch sizes is necessary to back the efficiency narrative.
5. Important engineering choices are fixed or weakly explored, which limits external validity. Hyperedges have a hard cap of 18 nodes, the construction follows a prior recipe, and γ is learned but only initialized to 1, with scant sensitivity analysis beyond the scale count T; moreover, training uses YFCC100M and then evaluates on multiple targets without a thorough study of cross-dataset generalization or overfitting to YFCC statistics.

**Questions:**

Please refer to the weakness.

---

### Official Review · Reviewer_dRzC · 2025-10-31

**Soundness:** 2
**Presentation:** 3
**Contribution:** 2
**Rating:** 6
**Confidence:** 3

**Summary:**

The authors propose a correspondence method to improve the image matching. They design two graph energy functions to capture higher-order constraints and subgraph structural similarity, respectively. Their experimental results show that the proposed method outperforms the existing approaches.

**Strengths:**

To capture higher-order constraints, the authors introduce a hypergraph-based intra-graph energy function modeling the node consistency within a subgraph. To align the structures across subgraphs, the authors adopt a graph kernel function to measure their similarity.

**Weaknesses:**

The primary contribution of this paper is the enhancement of node and edge features, particularly by extending neighbor regions using a hypergraph strategy. While more accurate graph features are undoubtedly beneficial for improving graph matching performance, this paper lacks a theoretical analysis of the relationship between the proposed discrete-degree-based optimization and the resultant improvement in matching accuracy.

**Questions:**

The feature optimization is driven by intra-graph and inter-graph energy, calculated as the discrete degree of nodes and edges according to Equations 3 and 5, respectively. Intuitively, this optimization may help filter out outliers from the initial graph matching results. However, a critical question remains: could this process also inadvertently eliminate some inliers?

---

### Note · Authors · 2025-11-25

I have read and agree with the venue's withdrawal policy on behalf of myself and my co-authors.